


# Future changes in flash flood frequency and magnitude over the European Alps

Mar J. Zander[1,2], Pety J. Viguurs[3], Frederiek C. Sperna Weiland[1], and Albrecht H. Weerts[1,2]

[1]Department of Inland Water Systems, Deltares, Delft, the Netherlands
[2]Hydrology and Quantitative Water Management Group, Wageningen University Research, Wageningen, the Netherlands
[3]TAUW Nederland, Amsterdam, the Netherlands

**Correspondence:** M.J. Zander (marjanne.zander@wur.nl)

**Abstract.** Flash Floods are damaging natural hazards which often occur in the European Alps. Precipitation patterns and intensity may change in a future climate affecting their occurrence and magnitude. For impact studies, flash floods can be difficult to simulate due the complex orography and limited extent & duration of the heavy rainfall events which trigger them. The new generation convection-permitting regional climate models (CP-RCMs) improve the representation of the intensity and

frequency of heavy precipitation. Therefore, this study combines such simulations with high-resolution distributed hydrological modelling to assess changes in flash flood frequency over the Alpine domain. We use output from a state-of-the-art CP-RCM to drive a high-resolution distributed hydrological wflow_sbm model covering most of the Alpine mountain range on an hourly resolution. First, the hydrological model was validated by comparing ERA5 driven simulation with streamflow observations from 130 stations (across Rhone, Rhine, Po, Adige and Danube basins). Second, a hourly wflow_sbm simulation driven by

a CP-RCM downscaled ERAInterim simulation was compared to databases of past flood events to evaluate if the model can accurately simulate flash floods and to determine a suitable threshold definition for flash flooding. Finally, simulations of the future climate RCP 8.5 for the end-of-century (2096-2105) and current climate (1998-2007) are compared for which the CP-RCM is driven by a Global Climate Model. The simulations are compared to assess if there are changes in flash flood frequency and magnitude using a threshold approach. Results show a similar flash flood frequency for autumn in the future, but a decrease

in summer. However, the future climate simulations indicate an increase in the flash flood severity in both summer and autumn leading to more severe flash flood impacts.

## 1  Introduction

Flash floods are sudden torrential floods, triggered by high-intensity mostly short-duration localized rainstorms (Gaume et al., 2009; Amponsah et al., 2018; Kuksina et al., 2017). Flash floods have a high mortality rate compared to other flood types and

lead to serious economic damage (Gaume et al., 2009; Jonkman, 2005; Marchi et al., 2010). They often occur in mountainous areas where the combination of orography triggered convection, small catchments, and steep slopes can lead to a rapid concentration of runoff (Barredo, 2007; Kuksina et al., 2017). Therefore they are a major natural hazard in the European Alps.

Observational records show increases in the intensity of extreme rainfall over the past decades, primarily for short sub-daily durations (Westra et al., 2014; Förster and Thiele, 2020). This intensification is expected to become more apparent in the future





when higher temperatures lead to an increase in the moisture-holding capacity of the atmosphere (the Clausius-Clapeyron relationship) (e.g. Lenderink and Van Meijgaard, 2008, 2010; Ban et al., 2014, 2015). The intensity of flash floods and thereby their impacts may increase. Therefore, there is a need to adapt, and projections of future flash flood behaviour can inform adaptation strategies (Gobiet et al., 2014).

Their limited spatial and temporal extent and the relative remoteness of where they occur hampers documentation on flash floods in mountainous areas, hindering regional studies on flash floods (Modrick and Georgakakos, 2015).

For future impact studies, high-resolution data would be needed. Recent developments have opened the door for regional-scale flash flood modelling studies: The increasing availability of high-resolution Earth observation data enables high resolution distributed hydrological modelling (e.g. Imhoff et al., 2020; Eilander et al., 2021). The publication of regional, observation-based flood databases like Paprotny et al. (2017) and Amponsah et al. (2018) enable the validation of flash flood model results. Furthermore, with increasing computer power, climate modellers can downscale parts of global climate models (GCMs) to the kilometre scale: convection-permitting regional climate models (CP-RCMs). This scale increase improves the simulation of orographic precipitation due to the higher surface resolution. It enables the explicit computation of deep convection, where coarser-resolution regional climate models (RCMs) rely on parameterisation schemes for convective processes. This is a substantial source of errors and uncertainties for precipitation extremes (Prein et al., 2013, 2015; Ban et al., 2014; Lucas-Picher et al., 2021). CP-RCMs can therefore enhance our understanding of changes in rainfall extremes as they can improve the hourly statistics and diurnal cycle of the modelled rainfall as well as the spatial patterns of rainfall fields (e.g. Prein et al., 2013; Ban et al., 2014; Kendon et al., 2017; Fosser et al., 2017; Ban et al., 2021; Fumière et al., 2020). See Lucas-Picher et al. (2021) for a recent review.

The computational cost, runtime, and produced data volume limit the application of CP-RCMs to event-based experiments (e.g. Hazeleger et al., 2015; Manola et al., 2017; Schaller et al., 2020; Hegdahl et al., 2020), or simulation periods of about a decade (e.g. Leutwyler et al., 2017; Coppola et al., 2020; Ban et al., 2021; Pichelli et al., 2021).

While several hydrological modelling studies have used RCMs like the EURO-CORDEX simulations (Jacob et al., 2014) for climate change impacts on flooding (e.g. Smiatek and Kunstmann, 2019; Brunner et al., 2019; Di Sante et al., 2021; Alfieri et al., 2015), the application of CP-RCMs in hydrological studies is novel.

Pioneering recent work has shown that combinations of CP-RCMs and hydrological impact models can be applied to gain new insights on local and regional changes in flood impacts with a changing climate (Kay et al., 2015; Reszler et al., 2018; Felder et al., 2018; Rudd et al., 2020; Schaller et al., 2020). Although Kay et al. (2015) showed that finer resolution CP-RCMs do not automatically lead to more reliable hydrological modelling for large-scale river flooding, they indicate that small flashy catchments may show different results. Schaller et al. (2020) show that a model chain consisting of a CP-RCM and a distributed hydrological model can reproduce extreme rainfall events and subsequent flooding over two of such flashy mountainous catchments in Norway. Furthermore, hydrological modelling at an hourly resolution is necessary to capture the peak streamflow due to the fast flood generating processes. Using a 'storyline approach,' the events were translated into an ensemble of plausible future events taking RCP 4.5 for the end-of-century scenario. Not all modelled events in the ensemble hit the studied catchments



(Schaller et al., 2020).


Felder et al. (2018) use an event-based modelling chain with CP-RCMs, hydrological and hydraulic models, and loss modelling to explore the feasibility of determining flood impacts from simulated extreme weather in a GCM to the building scale in the Alpine Aare catchment. Their study, however, limits itself to one Alpine catchment. While Rudd et al. (2020) takes a regional approach to surface water flooding in Southern England using CP-RCM simulations to drive a gridded hydrological

model. They found larger changes in precipitation than in surface runoff, indicating an added value of using hydrological modelling over purely applying thresholds on the simulated rainfall amounts in CP-RCMs (Rudd et al., 2020).

Reszler et al. (2018) use an ensemble of two RCMs at three spatial resolutions including at convection-permitting scale (ca. 3 km) as input for distributed hydrological modelling in three catchments in the Austrian Alps. They find one of the CP-RCMs outperforms other simulations on most flood statistics including the seasonality of floods. However, they conclude to finding

no clear added value of the CP-RCM simulations due to lacking realism in the temporal distribution of rainfall intensities at a sub-daily scale and/or total precipitation amount per rainfall event (Reszler et al., 2018).

In this research, we take a regional approach to simulating flash floods in the European Alps using ten-year transient CP-RCM simulations and high-resolution hydrological modelling (1km). The regional approach allows for using the high spatial resolution of the climate simulation data. We aim to validate the ability of the CP-RCM - hydrological model combination to

simulate flash floods and assess changes in their frequency and magnitude for the current and future climate. The hypothesis is that by using a form of trading space for time, given the relatively short simulation periods of 10 years available and using a large enough spatial domain, we can assess future flash flood frequency and magnitude changes over the European Alpine domain. The transient periods ensure that hydrological processes and conditions important to flash flood development, such as initial soil moisture content, are considered.

**2   Data and Method**

**2.1   Study Area**

The European Alps are a mountain range of 800 kilometres long and have an average width of 200 kilometres (Schär et al., 1998). The ridge height is about 2.5 kilometres. The European Alps are characterised by deeply incised valleys and extensive lowlands and have considerable topographic variability (Gobiet et al., 2014). The study area covers most of the European

Alpine mountain range encompassing many river basins; the Upper Rhone river until Geneva, the High Rhine & Alp Rhine until Basel, Adige to just South of Trento, and the Alpine tributaries to the Danube, Alpine part of the tributaries to the Po basin. The study domain falls within Italy, Switzerland, Liechtenstein, Austria, and France. The modelled domain contains many glaciers, lakes, hydropower dams and reservoirs. See Figure fig:studydomain for an overview of the domain. The annual precipitation is around 1400-1500 mm/year, with more precipitation in the West and South (Quaile, 2001). The Alpine rivers

have snow and glacier melt regimes with peak discharges in late spring and early summer (May, June, July) and low discharges in winter (Meile et al., 2011). Most flash floods in the European Alps occur in late summer and autumn. In France and Italy,

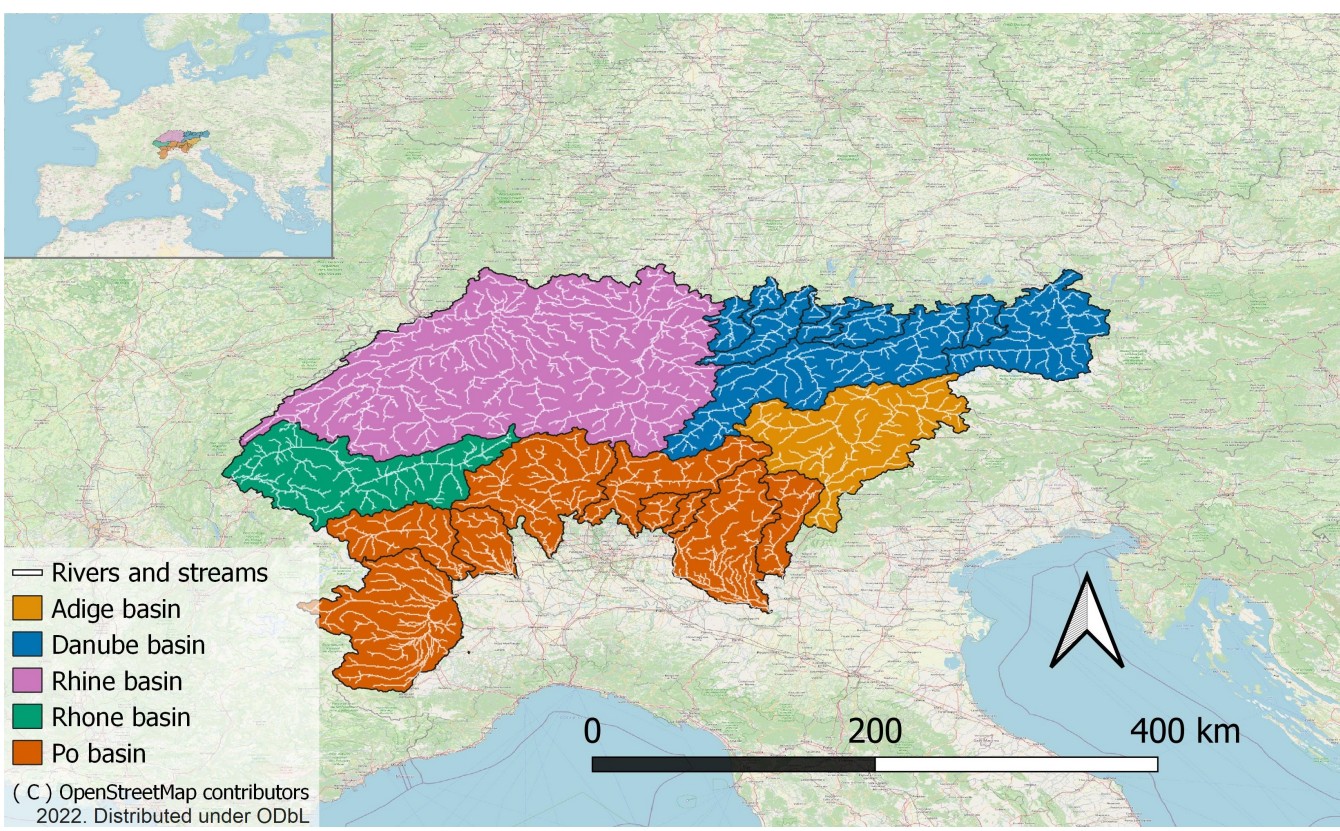

**Figure 1.** Hydrological modelling domain over the European Alps with the basins and tributaries to the Adige (yellow), Danube (blue), Po (orange), Rhine (pink) and Rhone (green), background map OpenStreetMap contributors (2017).

flash floods occur primarily in autumn (September, October, November), while in Austria, most flash floods occur in late summer (August, September) (Gaume et al., 2009). The high surface temperature of the Mediterranean Sea in late summer and autumn can load the atmosphere with large amounts of water vapour. Such loaded conditionally unstable air masses advect

to the coast, where the relief channels them and forces them to lift. This can lead to intense convective rainfall events, which cause flash floods (Doswell et al., 1996; Tarolli et al., 2012).

## 2.2    Convection Permitting Regional Climate Model

The convection-permitting climate model used is the Unified Model, MetOffice Hadley Center UKMO 2.2 km (UM version 10.1) as described by Berthou et al. (2018). In a recent CP-RCM intercomparison study, this model performed well in terms

of simulated precipitation over the Alps (Ban et al., 2021). Modelled precipitation is available as hourly sums, along with three-hourly near-surface air temperature, six-hourly surface pressure, and daily mean incoming solar shortwave radiation at the surface.





Three UM simulations are used: a reference UM simulation driven by Era-Interim reanalysis data as lateral boundary conditions (Dee et al., 2011) from 2000 to 2012, from hereon the 'Evaluation' simulation. And two climate simulations which are directly driven by the HadGEM3-GC3.1-N512 GCM and use a 360-day calendar (Williams et al., 2018). The current climate is simulated for the period of 1998-2007, referred to as the 'Historical Climate' simulation, and for the future climate change scenario RCP 8.5 van Vuuren et al. (2011) is used to simulate the period 2096 - 2105, from hereon 'Future Climate' (see Table table:simulations). The Evaluation simulation serves to assess the quality of the model, while the GCM-driven Historical Climate and Future Climate simulations serve to assess the differences that could be attributed to climate change.

### 2.2.1 Climate Model Data Processing

The climate model data is remapped from its rotated pole grid to WGS84 coordinates, remapping the precipitation conservatively. We remap all fields to the resolution of the hydrological model grid. We linearly interpolate the surface level pressure and temperature to hourly values. Because remapping daily mean radiation to an hourly sinusoidal solar radiation pattern is not unambiguous, we chose to assign a uniform radiation flux to each hour of the day corresponding to the modelled average. We expect this has minimal impact on the model outcomes; see also van Osnabrugge et al. (2019). From the temperature, pressure and radiation fields, the potential evapotranspiration is calculated using the formulation of de Bruin et al. (2016).

### 2.3 Hydrological Model

In this study, we use the wflow_sbm hydrological model (Schellekens et al., 2021; Imhoff et al., 2020), it is a spatially distributed bucket-style model. It uses the kinematic wave routing for surface, channel and lateral subsurface flow. The wflow_sbm is developed to maximise the use of high-resolution spatial data from Earth observations. Automated models can be set up for river basins around the globe using open data at various spatial resolutions. The model parameters are estimated from point-scale (pedo)transfer functions (Imhoff et al., 2020; **?**; **?**).

For this study, the domain is divided into seven submodels of 5 Alpine basins to improve computation times (Adige, Danube, Po, Rhine and Rhone, see Figure fig:studydomain). All models of the study domain are set up in the same manner with 0.008333-degree cell sizes, corresponding to roughly one kilometre. The river network is derived using the method of Eilander et al. (2021). In addition, we use the same a priori parameter estimation methodology as Imhoff et al. (2020).

We take the Hydro-MERIT high-resolution raster hydrography maps as the base for the modelling (Yamazaki et al., 2019). For the lakes, we use hydroLAKES (Messager et al., 2016), taking the minimum lake area to consider $1km^2$, we take reservoirs from the GRanD database likewise taking the minimum area to consider $1.0km^2$ (Lehner et al., 2011). We use monthly MODIS leaf area index for the vegetation and land use cover from CORINE (**?**European Environment Agency, 2012).

The Alpine glaciers are modelled using the Randolph Glacier Inventory dataset (Global Land Ice Measurements from Space (GLIMS), 2017), taking the minimum modelled glacier area to consider $0.1km^2$. We take the same initial glacier extents and volumes for each of the wflow_sbm simulations, so no scenario of glacier disappearances is considered (e.g. projected end-of-century volume loss of up to 98.8 % for RCP 8.5 in **?**). To test the sensitivity of our results to initial glacier extents and volumes, we simulate the Future Climate scenario for the Rhone catchment, the most glaciated of the modelled catchments,



with and without glaciers at the simulation start. No differences were found in the modelled flash flood occurrence and magnitude between the two simulations (see Supplement sec:supplement-rhone, Figure fig:supplement-rhone).

In wflow_sbm, the horizontal saturated hydraulic conductivity can be derived from the vertical hydraulic conductivity parameter. We use a constant multiplication factor of 100 for the whole modelled domain. In other studies, this factor was to be one of the most sensitive model parameters for the wflow_sbm model (Imhoff et al., 2020). Therefore, we performed a sensitivity analysis for the Alpine Rhine basin using ERA5 as forcing, in which this factor is varied from 5, 20, 50, 100, 200 & 500 [-] (see Supplement sec:supplement). The model performance is assessed for the observed discharge in the Rhine River at Basel and in the Thur River station at Andelfingen. We found that a factor of 100 was satisfactory, which is lower than the factor of 250 applied in Imhoff et al. (2020) for the entire Rhine basin up to the Netherlands. While the results were not very sensitive to the applied factor, the factor of 100 does show better performance for the Alpine Rhine at Basel and Thur (See Supplement  sec:supplement, Table table:ksathorfrac) For a more detailed overview of the wflow_sbm model setup, we refer to Imhoff et al. (2020).

### 2.4 ERA5 Validation simulation

First we force the wflow_sbm models with daily ERA5 reanalysis data (Hersbach et al., 2018), resampled to the hydrological model resolution, in order to assess the hydrological models' ability to simulate Alpine river discharges (Validation simulation, Table table:simulations). We validate the modelled discharge against streamflow observations from a total of 130 gauging stations. Six discharge timeseries are obtained from the Global Runoff Data Centre, seven from the Provincia Autonoma di Trento, 25 from the Bundesamt für Umwelt, and 92 from the LamaH dataset (Global Runoff Data Centre , 2021; Bundesamt für Umwelt, 2021; Dipartimento Protezione Civile, Provincia Autonoma di Trento, Trento, 2019; Klingler et al., 2021). These discharge time-series are not spread evenly over the study area, with more stations available in the Danube and Rhine basins than in the Southern and Western parts with the Rhone and Po basins. To assess the model performance, we calculate the Kling-Gupta Efficiency scores (KGE) (Gupta et al., 2009). As the temporal resolution of the observed discharge time-series differs, the KGE scores are calculated for mean daily discharges to make them comparable.

### 2.5 ERA-Interim Evaluation simulation

Next, we force the wflow_sbm models with the downscaled UM Era-Interim simulation data (Evaluation simulation, Table table:simulations). Doing so enables an assessment of the ability of the UM CP-RCM to simulate the synoptic situations leading to flash floods. We calculate the KGE scores for the same period of 2002 to 2012 for daily discharge values for the 130 gauging stations.





## 2.6 Flash flood validation

We use the HANZE database of damaging European Floods (Paprotny et al., 2017), and EuroMedeFF database of flash floods (Amponsah et al., 2018) to evaluate the ability of the climate and hydrological modelling chain to simulate flash floods. According to these datasets, nine flash floods took place within the study domain during the 2002 - 2012 period, see Table table:recordedfloods (Amponsah et al., 2018; Paprotny et al., 2017). Apart from the location, Amponsah et al. (2018) also lists the estimated peak specific discharge of the flash flood events.

To determine what will be considered a flash flood in our study, we use an adapted version of the definition of Amponsah et al. (2018):

– the affected catchment has a maximal size of $3000 km^2$ [3000]

– the peak specific discharge is at least $0.5 m^3 s^{-1} km^{-2}$ [0.5]3

Here the peak specific discharge is calculated by dividing the peak discharge by the upstream catchment area (Marchi et al., 2010). In this study, we calculate the modelled daily maximal specific discharges and compare these to the threshold value for the dates at which a flash flood happened. It is a well-known issue that internal variability in a climate model can cause rainfall events to be simulated at a different location and/or time compared to the driving data (e.g. Reszler et al., 2018; Schaller et al., 2020). Furthermore, the HanzeFloodlist provides information on the flood location at the level of municipalities, the European NUTS3 level. For these two regions, we take a regional approach to the flash flood validation. We consider it a validated flood event in the hydrological simulation if an event fitting the above mentioned criteria occurs within the submodel basin of the reported flash flood within three days of the reported date.

## 2.7 Comparison Current and Future flash flood occurrence

In order to assess potential changes in flash flood occurrence between the current and future climate scenario, we force the wflow_sbm models with GCM-driven CP-RCM simulations for 1998-2007 and 2096-2105 (Table table:simulations) and compare them. For the Future and Historical Climate simulations the specific discharge is calculated from the daily maximal discharge and upstream area for the summer (JJA) and autumn (SON) period, taking the first year of simulation as warm-up. The peak specific discharge threshold and maximal upstream area threshold are applied. We compare the frequency of threshold exceedances and the flood peak magnitudes (specific discharges) for the two simulations. Similar to Alfieri et al. (2015) we compare the two simulations after aggregation to the subbasin level as well as for the entire modelled domain (Adige, Danube, Po, Rhine and Rhone, see Figure fig:studydomain)





**Table 1.** Wflow_sbm Model simulations

| Simulation | Driving Data | Time Period | Source Spatial Resolution | Temporal Resolution |
|---|---|---|---|---|
| Validation | ERA-5 | 01/01/1979 - 31/12/2019 | $0.25°$ | daily |
| Evaluation | UM - ERA-Interim | 01/01/2000 – 31/12/2012 | $2.2km$ | hourly |
| Historical Climate | UM- HadGEM3-GC3.1-N512 | 01/01/1998 – 31/12/2007 | $2.2km$ | hourly |
| Future Climate | UM-HadGEM3-GC3.1-N512 RCP8.5 | 01/01/2096 – 31/12/2105 | $2.2km$ | hourly |

## 3 Results

### 3.1 ERA5 Validation simulation

As can be seen in Figure fig:obs-mod-hydrographs, the hydrological model can simulate the annual cycle of discharge with low flows in winter and snowmelt, leading to discharge peaks from May to July. For most stations the KGE score ranges from $0.4 - 0.7$ (figure fig:kge-hist), with a maximum of $0.82$ for the Felsenbach and Vent stations. The model shows performance at the Rhine at Basel of $KGE = 0.73$, and the Adige at Trento, $KGE = 0.56$, stations near their respective catchment outlets. There are no clear spatial patterns in the model performance (Figure fig:kge-map) and no dependency on station elevation or Strahler stream order (not shown). Some modelled headwaters have high KGE scores, like the Vermigliana in the Adige catchment ($KGE = 0.80$) and the Thur at Andelfingen ($KGE = 0.75$, Figure fig:obs-mod-thur-2004), while other headwater stations perform poorly. The latter is due to hydropower reservoirs and lakes which are not included in the hydrological model due to their limited size ($< 1km^2$). At both Klaushof station and Sausteinaste station, there are hydropower reservoirs upstream, which are not simulated, leading to very high overestimations of the streamflow ($KGE = -4.53, \beta = 5.8 \& KGE = -5.98, \beta = 5.3$ respectively). For the station at Galtür, Au Trisanna, two upstream lakes are also not modelled, leading to large overestimations ($KGE = -1.07, \beta = 2.4$).

### 3.2 ERA-Interim Evaluation simulation

For most stations, the ERA-Interim Evaluation UM driven simulation is in reasonable agreement with observations (Figure fig:obs-mod-hydrographs & Figure fig:kge-hist). However, the ERA-5 driven Validation simulation generally outperforms the Evaluation simulation, as can be seen in the histogram of station performance in Figure fig:kge-graph. As can be seen from the quantile-quantile plot in Figure fig:kge-scatter, the difference in performance increases for stations where the simulations perform poorer. In Fig fig:obs-mod-hydrographs

### 3.3 Flash Flood Validation

Table table:recordedfloods lists the nine recorded flash floods in the study area from 2002 to 2012. For all recorded flash floods, the modelled peak daily discharge was heightened compared to the preceding and following days. The modelled peak specific



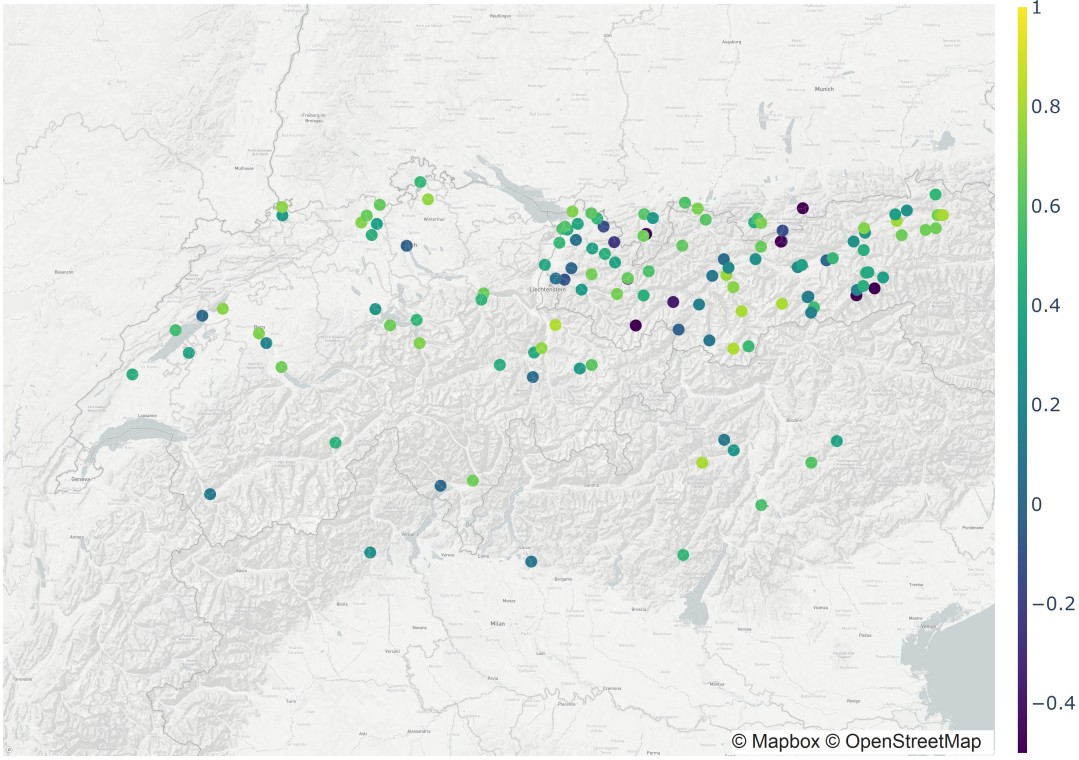

**Figure 2.** Map of the Kling-Gupta Efficiency score [-] for the 130 observation stations for daily discharges for 2002-2012 for the ERA5 Validation simulation.

discharge ranged from 0.47 to 2.4 $m^3 s^{-1} km^{-2}$ for the documented events. For the seven riverine flood events in the model area, the modelled peaks in specific discharge were notably higher than those modelled for the inundations classified as flash floods by Paprotny et al. (2017): ranging from 1 to 4.2 $m^3 s^{-1} km^{-2}$ (not shown). For the flash floods, the modelled peak specific discharges were all below the estimated ranges reported in Amponsah et al. (2018). We therefore deem the threshold of $0.5 m^3 s^{-1} km^{-2}$ suggested by Amponsah et al. (2018) as suitable.

### 3.4    Changes in Flash Flood Frequency and Magnitude

Figure fig:results-cdf-alps shows the cumulative distributions per modelled cell of the number of days on which the peak specific discharge exceeded the threshold of $0.5 m^3 s^{-1} km^{-2}$, from hereon threshold exceedances, for the Future and Historical Climate simulations and the summer and autumn seasons. Over the modelled domain, the summers have more threshold exceedances than autumns in both Historical Climate and Future Climate simulations. In summer, there are more days with

threshold exceedances in the Historical Climate scenario (Figure fig:results-cdf-alps ). At the same time, in autumn, we see hardly any difference between the number of days with threshold exceedances in the Future Climate scenario. However, there are some regional differences. The Future Climate autumn has more threshold exceedances than the Historical Climate for the





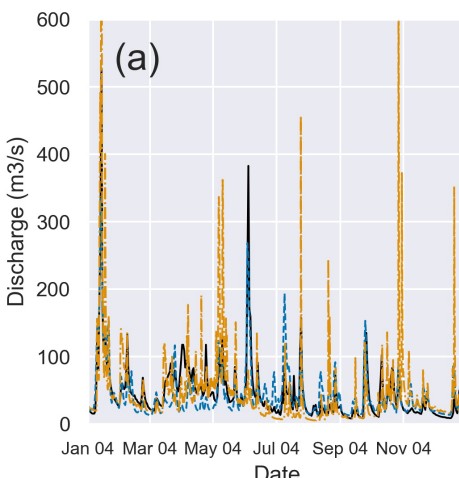

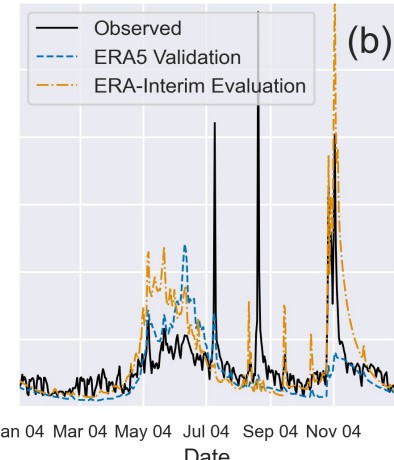

**Figure 3.** Observed discharge (black) compared to modelled discharges for the Validation ERA5 driven (blue) & Historical ERA-Interim driven (orange) simulations for the year 2004 for (a) Thur, Andelfingen station (Validation $KGE = 0.75$, Evaluation $KGE = 0.57$), and (b) Ticino, Bellinzona station (Validation $KGE = 0.67$, Evaluation $KGE = 0.21$).

**Table 2.** Recorded flash floods and simulated peak specific discharges in the ERA-Interim Evaluation driven simulation, using the EuroMed-eFF (Amponsah et al., 2018) and Hanze (Paprotny et al., 2017) flood databases.

| Start Recorded Date | Regions & Rivers | Modelled Peak Specific Discharge | Date Peak Specific Discharge | Source |
|---|---|---|---|---|
| 5-6-2002 | Sesia (Po) | 1.18 | 5-6-2002 | EuroMedeFF |
| 6-6-2002 | Rhone | 1.22 | 5-6-2002 | Hanze |
| 6-6-2002 | Danube, Rhine | 2.18 | 6-6-2002 | Hanze |
| 8-9-2002 | Rhone | 0.48 | 9-9-2002 | Hanze |
| 24-3-2005 | Rhine | 0.47 | 24-3-2005 | Hanze |
| 3-10-2006 | Isarco, Passirio (Adige) | 0.49 | 3-10-2006 | EuroMedeFF |
| 8-6-2007 | Zürich (Rhine) | 2.39 | 7-6-2007 | Hanze |
| 12-7-2008 | Po | 1.35 | 9-7-2008 | Hanze |
| 4-8-2012 | Vizze (Adige) | 0.66 | 6-8-2012 | EuroMedeFF |

Adige catchment. The seasonal differences and differences amongst the scenarios are small for the Alpine parts of the Po basin. While for the Rhine, the Historical Climate summer period has more threshold exceedances than the other periods. Summers have more threshold exceedances than autumns for the Danube basin but with slight differences between the two scenarios.

Figure fig:results-boxplot shows box-and-whisker plots of the magnitude of the daily maximal peak specific discharges for each of the threshold exceedances. Over the entire Alpine domain, both the Future Climate summer and autumn have a

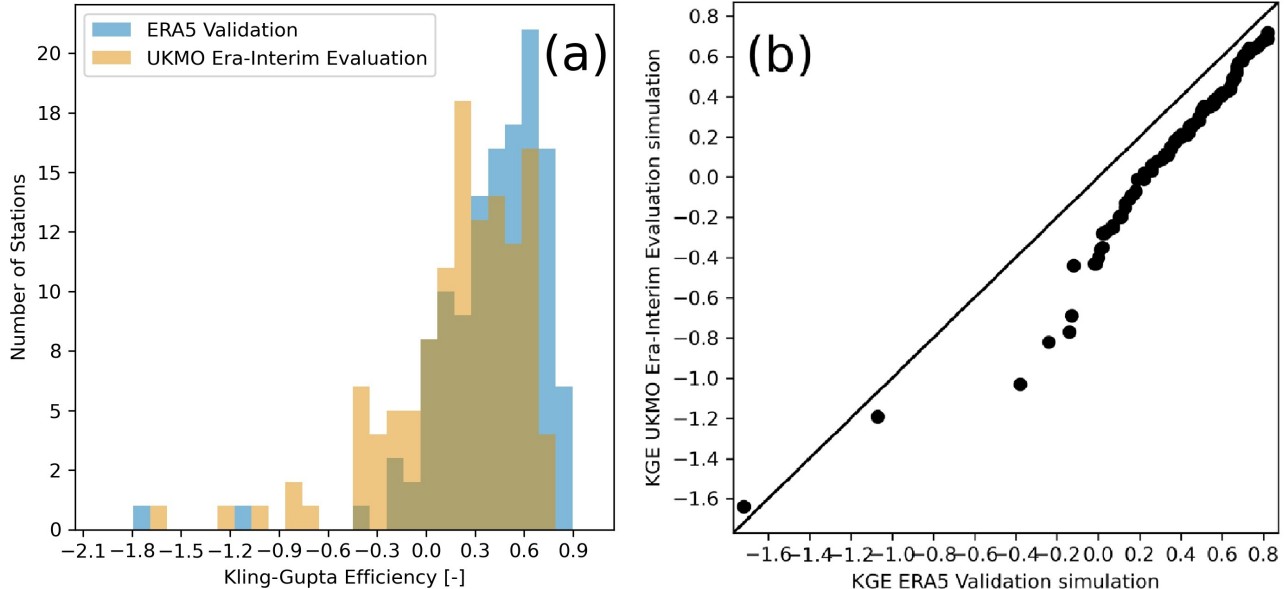

**Figure 4.** (a) Histograms and (b) Quantile-Quantile plot comparing the Kling Gupta Efficieny scores (KGE) [-] for the Validation ERA5 and ERA-Interim Evaluation simulations for the 130 stations.

higher 75 and 97.5th percentile of threshold exceedances, while the median magnitude is similar. The largest difference is in the maximal value in summer: $5.6m^3s^{-1}km^{-2}$ compared to $4.49m^3s^{-1}km^{-2}$ for the Historical Climate simulation. The

maximal peak specific discharge is the same in autumn for the two Climate simulations (4.26 vs 4.29). The difference between the two simulations is largest for the Adige basin, where the 50th 75, 97.5 and maximal values are higher in the Future Climate in both summer and autumn. In autumn the difference in maximal modelled specific discharge is $0.88m^3s^{-1}km^{-2}$ compared to $4.28m^3s^{-1}km^{-2}$ – a near fivefold increase.

In autumn, there are hardly any differences in the number of flash floods over the entire Alpine domain between the Current

and Future Climate. However, some basins will have threshold exceedances with much higher magnitudes (Adige, Po, Rhine), while others show similar maximum magnitude (Danube, Rhone).

The simulated decrease in future flash flood occurrence is most likely due to projected drying of the region, with higher temperatures, potential evaporation, and projected decreases in mean precipitation Gobiet et al. (2014); Ban et al. (2015). The antecedent soil moisture can play a significant role in the catchment response to extreme rainfall (Gaume et al., 2009). These

effects will be less strong in autumn. We explicitly account for these effects by simulating these hydrological conditions over the ten-year time slices.



**Figure 5.** Cumulative distribution of the number of days on which the threshold of $0.5 m^3 s^{-1} km^{-2}$ peak specific discharge is reached over the modelled domain in summer (JJA, solid lines) and autumn (SON, dashed lines) for the Current Climate (green) and Future Climate simulation (orange). The subpanels show the entire Alpine domain, Adige, Danube, Po, Rhine and Rhone basins (see Figure fig:studydomain).





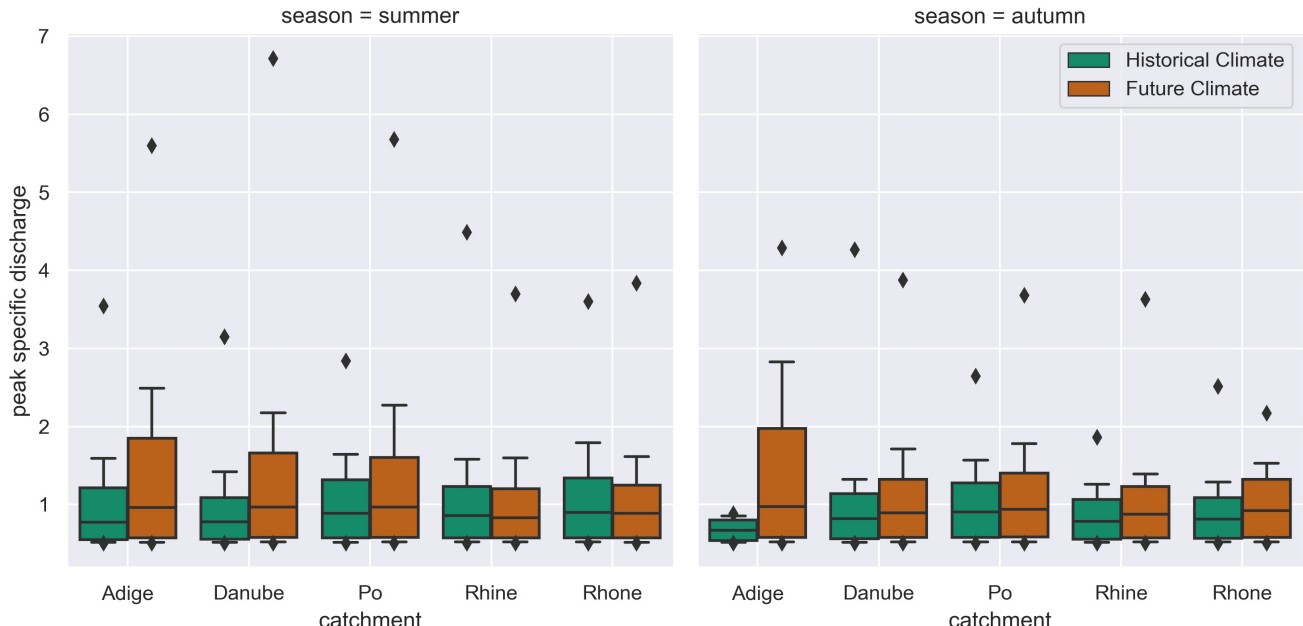

**Figure 6.** Box-and-whisker plots for maximal daily specific discharge exceeding the 0.5 $m^3 s^{-1} km^{-2}$ specific threshold for summer (JJA) and autumn (SON) for the Historical Climate and Future Climate simulation. The boxplots show the $25^{th}$, $50^{th}$ & $75^{th}$ percentile, with whiskers for the $2.5^{th}$ & $97.5^{th}$ percentile and diamonds for the maximal value.

## 4    Discussion on the methodology

Periods of 10 years are used in this study, as the CP-RCM model calculation time, data storage, and network requirements inhibit using longer periods (Prein et al., 2015). With such time slices, it is possible to move beyond event-based climate
impact studies (e.g. Felder et al., 2018; Schaller et al., 2020). However, deriving local statistics of change is hampered. Similar to Alfieri et al. (2015) and Rudd et al. (2020), the study presented here, therefore, takes a regional approach, aggregating results over larger regions. Just as in Rudd et al. (2020) we do not attempt to estimate return periods of extreme events as the simulation periods are too short to warrant such an approach. Recently, alternative statistical methods such as 'metastatistical extreme value analysis' have been applied to derive extreme value statistics from shorter time series (e.g. Marani and Ignaccolo,
2015; Zorzetto et al., 2016; Marra et al., 2018). However, these methods are applied to rainfall extremes and not the subsequent hydrological response (e.g. Li et al., 2017).

For both climate and hydrological modelling, we only consider changes due to a different radiation balance by using scenario RCP 8.5. All simulations use the same initial conditions for the landscape characteristics in the hydrological model. We do not consider any scenarios of 21st-century land-use changes or plant adaptation to changes in $CO_2$ concentration. This limitation
is familiar to many hydrological climate change impact studies (e.g. Alfieri et al., 2015; Brunner et al., 2019).





The CP-RCM both in the GCM driven and ERA-interim driven simulations contain biases. For instance, the UM has a wet bias in daily mean precipitation above 800 m elevation (Berthou et al., 2018; Ban et al., 2021). It was explicitly chosen not to apply a bias correction, downscaling, or a delta change approach to the climate model data as these techniques can disturb the change signal (Hagemann et al., 2011; Themeßl et al., 2012; Cloke et al., 2013; Reszler et al., 2018). Additionally, no

homogeneous datasets exist for bias correction for the entire modelled Alpine domain at the resolution and time-step of the CP-RCM data (Ban et al., 2021). By remapping ('downscaling') the climate data to the model resolution, we maintain the spatial coherence of the modelled precipitation and temperature fields. Furthermore, because we directly compare the two GCM driven climate simulations and thus focus on relative changes, the importance of biases for quantifying changes in flash flood frequency and magnitude is lessened.

The hydrological simulation which directly uses ERA5 reanalysis data outperforms the simulation which uses the ERA-Interim UM CPM data as lateral boundary forcing (see section sec:results-era5-eraint). This can be attributed to both the quality improvement from the Era-Interim to the ERA5 reanalysis product as well as the internal variability introduced to the CP-RCM simulation by only forcing it at the boundaries (Hersbach et al., 2018; Nogueira, 2020; Lavin-Gullon et al., 2021). The ERA 5 reanalysis dataset was, however not yet available when the UM modelling experiment was set up. Therefore, we

expect more minor differences in the model performance (here KGE) for a UM simulation using ERA5 as boundary conditions and a good agreement with observed discharges.

## 5   Conclusions and Outlook

This study presents a first regional modelling approach to studying future changes in flash flood occurrence and magnitudes. It uses a single high-resolution convection-permitting regional climate model as input for a distributed high-resolution hydro-

logical model. Other studies have focused on either smaller catchments (e.g. Felder et al., 2018; Reszler et al., 2018; Schaller et al., 2020), only considered surface water flooding (Rudd et al., 2020), or only focussed on riverine flooding using coarser resolution climate simulation data for flood changes studies (e.g. Smiatek and Kunstmann, 2019; Brunner et al., 2019; Di Sante et al., 2021; Alfieri et al., 2015).

The wflow_sbm model shows satisfactory performance, and the modelling chain of CP-RCM and distributed hydrological

modelling can reproduce recorded flash floods in the 2002-2012 period. Similar to the work of Rudd et al. (2020), we show added benefit of using the combination of convection-permitting climate model and hydrological modelling as changes in precipitation do not translate one-on-one into changes in flash floods.

Using CP-RCM simulations driven by a GCM of the Historical Climate and the end-of-century RCP 8.5 scenario as input to the distributed hydrological modelling, the frequency of occurrence and the magnitude of the specific peak discharge are

compared. The frequency of flash floods stays the same in autumn, with more severe extremes in some basins. In summer, we find a decrease in the frequency of flash floods, but with more severe extremes. This difference in maximal peak specific discharge is most distinct in summer, where the highest simulated peak discharges are higher in the future for all basins except



the Rhine. The Adige catchment shows the largest difference in simulated flash flood magnitudes between the Future and Historical Climate.

Assuming a relation between the flood peak and the flood impacts, we speculate that although the number of flash floods will decrease, they will become more devastating.

The construction of climate models always involves a simplified presentation of real-world processes and the need for parameterizations. To account for the uncertainties, we plan to extend this analysis in future work with an assessment based on multiple CP-RCMs.

*Author contributions.*   Marjanne Zander: Methodology, Validation, Formal analysis, Investigation, Data curation, Writing - original draft, Visualization.

Pety Viguurs: Methodology, Validation, Formal analysis, Investigation, Writing - review & editing.

Frederiek Sperna Weiland: Conceptualization, Methodology, Writing - review & editing, Supervision.

Albrecht Weerts: Conceptualization, Methodology, Writing - review & editing, Project administration, Funding acquisition, Supervision.

*Competing interests.*  Prof. dr. ir. A.H. Weerts is editor at HESS.

*Acknowledgements.*  This work was conducted as part of and supported by the EU Horizon 2020 Programme for Research and Innovation under grant no. 776613 (EUCP: European Climate Prediction system). We thank our Deltares colleagues Christian Liguori for the help with the sensitivity analysis and Dirk Eilander for the help in setting up the hydrological models. From the MetOffice, we thank Steven Chan and Ségolène Berthou for their help in providing and processing the climate data. From the Netherlands EScience Center, we thank Evert Rol
and Peter Kalverla for their help in providing the platform for data processing.





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

## Appendix A: Hydrological model parameter sensitivity analysis

**Table A1.** Sensitivity analysis of Horizontal Saturated Hydraulic Conductivity as a factor of the Vertical Saturated Hydraulic Conductivity (KsatHor) for discharge at the Rhine river at Basel and the Thur river at Andelfingen.

| Parameter | Rhine basin submodel (2002-01-01 - 2010-12-31) - Mean Daily Discharge | | | | | | | | | | | |
| --- | --- | --- | --- | --- | --- | --- | --- | --- | --- | --- | --- | --- |
| | Basel | | | | | | Thur - Andelfingen | | | | | |
| KsatHor | 5 | 20 | 50 | 100 | 200 | 500 | 5 | 20 | 50 | 100 | 200 | 500 |
| RMSE | 299.47 | 303.34 | 291.79 | 277.67 | 264.79 | 257.72 | 45.63 | 42.09 | 37.94 | 34.40 | 31.94 | 32.07 |
| NSE | 0.47 | 0.45 | 0.49 | 0.54 | 0.58 | 0.61 | 0.13 | 0.26 | 0.40 | 0.51 | 0.57 | 0.57 |
| KGE | 0.74 | 0.73 | 0.73 | 0.72 | 0.69 | 0.63 | 0.61 | 0.65 | 0.70 | 0.75 | 0.75 | 0.58 |
| cc | 0.75 | 0.73 | 0.74 | 0.75 | 0.77 | 0.79 | 0.77 | 0.77 | 0.77 | 0.78 | 0.78 | 0.76 |
| alpha | 0.94 | 0.99 | 1.05 | 1.12 | 1.20 | 1.30 | 0.69 | 0.74 | 0.82 | 0.93 | 1.08 | 1.33 |
| beta | 0.99 | 0.98 | 0.97 | 0.96 | 0.96 | 0.95 | 0.93 | 0.93 | 0.92 | 0.92 | 0.91 | 0.90 |

## Appendix B: Sensitivity of model results to end-of-century glacier retreat for the Rhone basin submodel

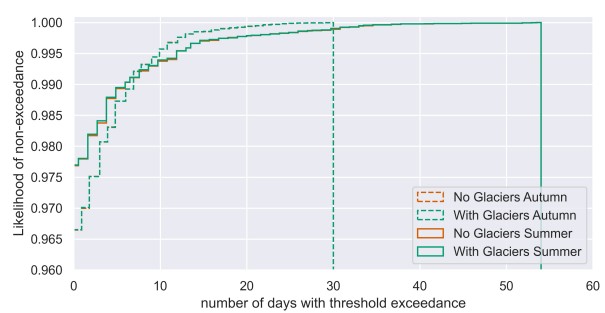

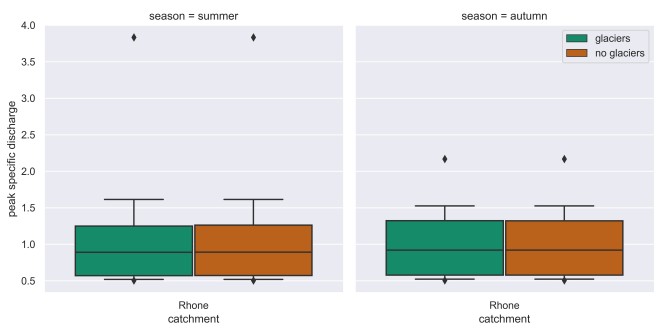

(a) Threshold exceedance cummulative frequency distribution, same methodology as figure fig:results-cdf-alps

(b) Threshold exceedance magnitudes, same methodology as figure fig:results-boxplot

**Figure A1.** Comparison Future Climate simulation for summer (JJA) and autumn (SON) for the Rhone with initialised glacier extents and volumes ('glaciers', green) and without glaciers to reflect a scenario with complete glacier retreat for the end-of-century ('no glaciers', orange).