# Peer review of "Future changes in flash flood frequency and magnitude over the European Alps"

_Hydrology and Earth System Sciences, 2022_

## Author Comment (AC2)

Authors response to Anonymous Referee #2

We thank the referee for their time and effort in reviewing our manuscript.

Below we address your comments and say how we will incorporate the feedback into an improved manuscript. Author responses are in blue.

This study used CP-RCMs and hydrological model to study the flash flood frequency over the Alpine domain. The topic is interesting. I have some major comments shown below:

1. Although this study focuses on the flash flood changes in the future climate, it is necessary to show the changes in annual precipitation and heavy precipitation between future climate and current climate conditions in the study area. These changes can be compared with the flood changes.

    Pichelli et al. 2021 and Chan et al. 2020 compare the simulated precipitation of the CP-RCM simulations for the current and future climate (RCP8.5), which we found to be sufficient for the purposes of our study. The results of these evaluations will be more elaborately discussed in the revised manuscript in Section 2.2.

    Rudd et al. (2020) found that changes in threshold exceedances were larger for simulated future precipitation (RCP 8.5), when compared to changes in discharge threshold exceedances when the CP-RCM simulations were used to drive a processed-based hydrological model. We deemed a replication of the work of Rudd et al. (2020) outside of the scope of our study, we will elaborate the change in precipitation over the study domain between the historical and future climate scenario in section 2.2 and the discussion.

2. What are the changes in snowmelt dominated flood in the future climate? In a warmer climate, it is expected to have significant changes in snow melt dominated flood.

    We have limited the scope of our study to the summer and autumn seasons as convective events are dominant in these seasons, so using convective-permitting climate simulations would provide most added benefit over coarser simulations of future climate scenarios (e.g. Pichelli et al., 2021, Lucas-Picher et

al., 2021). Snow and glacier melt dynamics play a lesser role than in (late) spring.

3. What are the differences between the changes in hourly and daily flood frequency and peaks?

   For data processing considerations, in determining if the threshold is exceeded, we take the maximal hourly discharge of each day. We consider this the most relevant for our analysis. We consider changes is hourly and daily flood frequency out of scope. We will clarify the text in section 2.6 on this.

4. What are the differences in air temperature between the current climate and future climate conditions in the study area?

   We agree with the reviewer that this is a current lack in the literature as the analyses of convection-permitting regional climate models have focused on the simulation of precipitation. We will elaborate the manuscript here.

5. The major parameters of the hydrological models should be shown in the manuscript. What are the differences of parameters used in different catchment? Are there any parameters related to snowmelt or glacier melt?

   For the sake of brevity, we have kept the description of the model parameters concise and referred to Imhoff et al. 2020. We will elaborate the model description. More details on the model and the processes considered can be found in Van Verseveld et al., 2022.

   Glaciers are modelled with the two main processes: glacier build-up from the conversion from snow to ice, which is modelled with two parameters: threshold temperature for which precipitation falls as snow on a glacier and a snow-to-ice conversion fraction per timestep; and glacier melt using a degree-day model. Likewise, snowmelt is simulated with a degree-day approach (Imhoff et al., 2020, Van Verseveld et al., 2022 in review)

6. Figure 2, it is better to add the boundary of catchment on the figure.

   We thank the reviewer for the suggestion, we will do so.

7. Figure 3, it is better to add precipitation on the figure to compare with the simulated and observed flood.

   We thank the reviewer for the suggestion, we will consider doing so.

8. It is better to add more descriptions of RCM models, for example, the parameter scheme for convection and precipitation simulation.

Deep convection is resolved rather than parameterized in the used CP-RCM simulation. We chose to refer to the papers in which the used simulations are described for the descriptions of the parameterization schemes and dynamical cores used for the CP-RCM simulations.

References:

Steven C Chan, Elizabeth J Kendon, Ségolène Berthou, Giorgia Fosser, Elizabeth Lewis, and Hayley J Fowler. "Europe-wide precipitation projections at convection permitting scale with the unified model". *Climate Dynamics,* 55(3):409–428, 2020. https://doi.org/10.1007/s00382-020-05192-8

R.O. Imhoff, W.J. van Verseveld, B. van Osnabrugge, and A.H. Weerts. "Scaling point-scale (pedo)transfer functions to seamless large-domain parameter estimates for high-resolution distributed
hydrologic modeling: An example for the Rhine river". *Water Resources Research*, 56(4), April 2020. https://doi.org/10.1029/2019WR026807

Emanuela Pichelli, Erika Coppola, Stefan Sobolowski, …, Jesus Vergara-Temprado. "The first multi-model ensemble of regional climate simulations at kilometer-scale resolution part 2: historical and future simulations of precipitation." *Climate Dynamics*, 56(11):3581–3602, 2021 https://doi.org/10.1007/s00382-021-05657-4

W. J. van Verseveld, A. H. Weerts, M. Visser, J. Buitink, R. O. Imhoff, H. Boisgontier, L. Bouaziz, D. Eilander, M. Hegnauer, C. ten Velden, and B. Russell. "Wflow sbm v0.6.1, a spatially distributed hydrologic model: from global data to local applications". *Geoscientific Model Development Discussions*, 2022:1–52, 2022 (in review) https://doi.org/10.5194/gmd-2022-182

---

## Author Comment (AC3)

Authors response to Anonymous Referee #1

We thank the reviewer for the time and effort taken for reviewing our manuscript.

Below, we address the comments and how we will improve our manuscript based on the feedback. Authors responses are in blue.

As becomes clear below, I do not see evidence that the modelling framework is fit to model flash flood frequency. It rather seems like the model applies some filtering to the precipitation events but we do not know how realistically this generates flash floods. Accordingly, the added value of the hydrological model as compared to frequency analysis on the precipitation directly is not convincingly demonstrated or discussed.

Reply:

We do not see the modelling chain to be detached from the flood generating mechanisms. We use a process-based distributed hydrological model that accounts for antecedent soil moisture conditions and ensures the spatial aggregation of rainfall to runoff and flash floods over larger spatial extents. The fast flood generation of flash floods requires a modelling framework which is able to simulate with short time steps and a high spatial resolution (e.g. Schaller et al., 2020) the wflow_sbm model simulates processes that are important for flash flooding (e.g. Infiltration excess, saturation excess, Hortonian overland flow, see Verseveld et al (2022)).  The model output variable we consider is the surface runoff / overland flow which represents, once the amount crosses the flash flood threshold, the occurrence of flash floods. We will make this more clear in both the introduction and methodology to clarify our motivation of using this hydrological model.

Regarding the precipitation analysis:

Line 63-66 of the Introduction discuss the work of Rudd et al. (2020) in which CP-RCM data was used to drive the distributed process-based hydrological model Grid2Grid. Rudd et al. (2020) performed threshold analyses on both the CP-RCM simulated precipitation and the Grid2Grid simulated discharge. They found smaller changes in threshold exceedances between the current climate and an end-of-century future climate scenario for discharge than for precipitation. Rudd et al. (2020) thereby demonstrated the added value of processed-based hydrological modelling over precipitation threshold analyses for hydrological impact assessments. The hydrological model introduces a buffering capacity. We therefore deem that including a hydrological model in the modelling chain when researching flash floods is valid. We deem a replication of the work of Rudd et al. (2020) outside of the scope of our study.

The simulated precipitation of the CP-RCM has been evaluated in previous work (e.g. wet day frequency, diurnal rainfall intensity, rainfall intensity, and peak hourly rainfall intensity) (Berthou et al., 2018, Chan et al., 2020, Ban et al., 2021, Pichelli et al., 2021), which we found to be sufficient for the purposes of our study. The results of these evaluations will be more elaborately discussed in the revised manuscript in Section 2.2.

This problem is enhanced by the fact that according to the text (methods section), only 9 flash floods are contained in the used data base for the entire region and study period. For these floods, the model validation is summarized as: "For all recorded flash floods, the modelled peak daily discharge was heightened compared to the preceding and following days." (Section 3.3). This seems a weak argument.  The regional approach to flash flood validation (an observed event is validated if something happened within the three days in

one of the 5 large subbasins, Fig. 1) is also not convincing: in as far does this demonstrate that the hydrological model has any added value compared to the precipitation input?

Concerning the validation for flash flood modelling: To the best of our knowledge, we used the available scientific databases of recorded floods: Paprotny et al. (2017) and Amponsah et al. (2018). Our choice of using both Paprotny et al. (2017) and Amponsah et al. (2018) is to increase the sample size. The number of flash floods which are reported in the database for the time-period considered is limited even though we have selected quite a large mountainous study area. Unfortunately, the limited time-series length of available of CP-RCM data does not allow for extending the time-period to cover more flashflood. During the studied period the small amount of flash flood is not unexpected.

Therefore, in addition to these recorded flash floods we use discharge data to validate the modelling chain. Concerning the overall hydrological model performance, we show in Figure 2 the Kling-Gupta Efficiency score. The KGE values indicate that the model forms a reasonable representation of the hydrological processes within the basin. As noted previously, we deem the further evaluation of the precipitation fields is out of scope for this study.

Concerning the regional approach: The text will be clarified in this regard. The ERA-Interim CP-RCM simulation is only forced by ERA-Interim at the regional climate model boundaries. This can lead to internal variability which can lead to precipitation events being modelled at the wrong location (e.g. Schaller et al 2020). The regional approach does not strongly penalize this spatial difference.

Additionally, we will add an analysis of the peaks in the observed discharge compared to the modelled discharge for threshold exceedances. From confusion matrices between the observed and modelled discharge at the observation stations we will show the Peirce Skill Score. This analysis does penalize deviations in the modelled location of precipitation events.

Furthermore I do not understand how the flash flood producing threshold is validated, i.e. I do not understand what is actually validated (lines 215-220). This threshold is however the key to judge if flash flood frequencies increases or decreases.

We agree lines 215-220 are not clear.

An additional analysis will be included showing the Peirce Skill Score of the threshold exceedances for the discharge stations at which the threshold specific discharge of 0.5 $m^3s^{-1}km^{-2}$ after is exceeded in the ERA-Interim simulation. The Peirce skill score compares the observed and simulated discharges in terms of threshold occurrence and non-occurrence, penalizing false negatives more heavily than false positives (the unfulfilled simulations of threshold occurrences).

Finally: the study does not work with downscaling or bias correction (see discussion: "it was explicitly chosen not to apply a bias correction, downscaling, or a delta change approach to the climate model data as these techniques can disturb the change signal"). I have a hard time to understand how the output of the hydrological model is of any use in this case, in particular because it aggregates model outputs from catchments for which the bias of precip., temperature or their variability might be important but spatially different. This problem is not reduced by comparing similar simulations for the reference and the future period since flash flood analysis corresponds to the analysis of extreme events that might be crucially influenced by biases, and differently under current climate compared to future climate.

Thanks for this comment. Of course we downscaled the CP-RCM data (and ERA5) to the Wflow_sbm model grid and while doing this we applied downscaling using lapse rate correction (using the DEM). We will correct this in the new manuscript. There are currently no observational datasets available to bias correct the CP-RCM simulation data on the necessary high spatial and temporal resolution (Ban et al., 2021) for the entire modelled domain. In a revision of the manuscript, we will add to the discussion how limitations on fine-scale information from gridded observational precipitation datasets over mountainous terrain can cause issues when using as basis of bias correction (Lundquist et al., 2019 and Berg et al., 2015 according to Lucas-Picher et al., 2021).

Furthermore, the same studies have demonstrated that bias correction of temperature or precipitation is not a trivial exercise, it can change the spatial dynamics and disturb the change signal (line 264).

Detailed comments:

- Introduction: I would suggest to add references / arguments for "The intensity of flash floods and thereby their impacts may increase". Is precipitation intensity the only driver? What is the role of soil moisture, infiltration capacity? Is looking at precipitation intensity sufficient or would you need to consider compound events of high precipitation intensity and low antecedent moisture? I do not mean to suggest that you need to address all this but it would be good to give the bigger picture and to not oversimplify the case of flash floods; we also would need to have information (in the methods part) if the chosen model can reliably reproduce Hortonian and saturation-excess floods and who this is validated for the selected catchments

  As mentioned above we use a processed based hydrological model and simulation periods of 10 years to account for antecedent moisture content. More details on the model and the processes considered can be found in Van Verseveld et al., 2022. As mentioned before, we will make this clearer in the introduction and methodology section.

- Introduction: the sentence "Although Kay et al. (2015) showed that finer resolution CP-RCMs (..) for large-scale river flooding," is surprising, you just said before that using CP-RCMs for hydrological impact modelling is novel
  We will rephrase to "not common practice". To the best of our knowledge, we have incorporated all publications using convection-permitting regional climate models for hydrological studies which were published at the time of submission.
- Line 56 following: unclear if you talk about your own study or about the study in Norway?
  Line 54 to 59 all refer to the work of Schaller et al. (2020), this will be clarified by rephrasing "Furthermore the authors conclude that".
- Introduction: "However, they conclude to finding no clear added value of the CP-RCM simulations due to lacking realism in the temporal distribution of rainfall intensities at a sub-daily scale and/or total precipitation amount per rainfall event (Reszler et al., 2018)." Did you check in your work that the used precip series (without downscaling) have a realistic temporal distribution?
  Such evaluations have indeed been performed: Berthou et al. (2018), Ban et al. (2021) and Pichelli et al. (2021) evaluate the diurnal cycle of the modelled CP-RCM precipitation. These analyses only cover the parts of the simulation domain for which high-resolution hourly gridded precipitation datasets were available.

We will include a paragraph about the precipitation evaluation in the methodology section.
The modelled peak precipitation is in mid-late afternoon, which is an improvement compared to coarser RCMs. In the Swiss Alps, the timing of the peak precipitation is too early, between 2 PM and 4 PM, compared to observations 4 PM to 8 PM (Berthou et al., 2018).

- Study area: a map with mean precip properties (annual totals, intensity of e.g. 1-day precip events) would complete the picture; with the presented information, we have no idea how spatially variable the meteorological drivers are
  We thank the reviewer for the suggestion. More insight into the observed and modelled precipitation will be useful for the reader and we will add the suggested or alike maps.
- Study area: "Most flash floods in the European Alps occur in late summer and autumn"; do you have a reference? later on, "in France xxxx" : these sentences are misleading since this analysis is only about a specific part of France, Italy etc. Do you have detailed information on flash floods in the catchments that you actually include in the study? What about flash floods in the Northern parts (Aare / Rhine catchments); are there many?
  Yes, the reference Gaume et al, 2009 is given in line 93.
  For Switzerland, summer is the most important season for floods including flash floods (Froidevaux et al., 2015).
- Hydro model: could you perhaps mention the number of model parameters that are estimated / assigned following "same a priori parameter estimation methodology as Imhoff et al."? which parameters potentially influence flash floods the most? Does the model have Hortonian and saturation-excess surface runoff?
  The model simulates both infiltration-excess and saturation-excess surface runoff (Van Verseveld et al., 2022 in review).
- Can you elaborate on why the vertical hydraulic conductivity parameter is the most sensitive one? And is this result from previous work transferable to here? I guess the sensitivity of wflow parameters depends on what hydrological time scale is dominant for the considered catchment / scale: either processes leading to water mobilization at the hillslope scale or processes of lateral transfer (surface and subsurface) to the stream network or routing within the stream network; if you only assess hydraulic conductivity you implicitly assume that water mobilization is dominant? But what about lateral flow to the stream network?
  The reviewer might have miss read the manuscript here. One of the most sensitive parameter is the lateral hydraulic conductivity (horizontal hydraulic conductivity). This is already mentioned in the manuscript. In the supplement an analysis of the effect of this parameter on modelled results is given. Note that for the lateral conductivity not good transfer function is available and we use the same approach as Bell et al (2007) who assumed that the lateral hydraulic conductivity is equal to the vertical hydraulic conductivity times a factor (see also to Sperna Weiland et al (2021), Wannasin et al (2021), Aerts et al (2022) and Verseveld et al., 2022).

- What is a "constant multiplication factor of 100"? how does it influence hydraulic conductivity?
  The multiplication factor [-] between the vertical and horizontal saturated hydrological conductivity. The vertical saturated hydraulic conductivity is multiplied by this factor to estimate the horizontal saturated hydraulic conductivity in order to

calculate lateral subsurface flow. The vertical hydraulic conductivity is estimated based on (pedo)transfer function (see Imhoff et al 2020).

- Why did you use ERA5 for the sensitivity study and not downscaled ERAinterim? Should you not test the model sensitivity with the type of input data that you use thereafter? What is the added value of the validation with ERA5? If the model performs better with ERA5, how is this useful for the final aim, generating flash floods under future climates?

  The ERA5 datasets is a follow-up of the ERA-Interim dataset and therefore assumed to perform better than the ERA-Interim dataset. It is therefore assumed that using the ERA5 datasets we will obtain better model performance with more realistic parameterization and we aim for the best model set-up. Performing a sensitivity analysis with ERA-Interim data may result in parameterization compensating effects for input data biases. Unfortunately, the CP-RCM simulations have only been performed with the ERA-Interim datasets as boundary conditions. They were conducted at an earlier stage. We also run the Wflow_sbm model with the downscaled ERAinterim forcing and the comparison of the hydrological model between both simulations is shown (Figure 4).

- What was the time step of ERA5 combined with wflow? Hourly and daily, this is unclear? What is the value of daily model performance assessment for a model that is later on used to derive flash floods? I miss some convincing arguments that the analysis framework is actually able to reproduce extreme streamflow events of the flash flood type.
  The ERA5 simulation where conducted with daily rainfall (Table 1). The simulation with the downscaled ERAinterim (and current and future climate) simulated forcing data where conducted with an hourly timestep. We compared both the daily and hourly simulations against streamflow.

- Is the quality of the modelling chain with CP-RCM driven by a Global Climate Model assessed in any way? E.g. in terms of the flash flood frequency for present day? Otherwise, how can you justify that this model chain gives valuable results?
  Please see our comments to one of the earlier remarks where we refer to the evaluation against flash flood occurrences from the databases of Paprotny et al. 2017 and Amponsah et al. 2018. This is the first step in the validation where we show that the model is able to simulate flash floods and in the second step KGE values have been calculated for the wflow_sbm forced with both the ERA5 and CP-RCM datasets. For the validation of the CP-RCM precipitation dataset by itself we refer to Chan et al. 2020, Ban et al. 2021.

- How are lakes treated in the model? Does flash flood analysis below large lakes make any sense?
  Lakes are included in the model above a certain area threshold ($1km^2$). For the flash flood analysis we only look at catchments <3000km2 and a flash flood threshold of $0.5\ m^3s^{-1}km^{-2}$. We agree that below a large lake this makes less sense and we will perform an additional check on this.

- The paper would highly benefit from a sketch that summarizes the work scheme (what data went into what model and how the output was assessed / used)? Instead of Table 1, which does not mention to what the model output is compared to
  We thank the reviewer for the suggestion.

- The focus of the paper is on the European mountain range where snowmelt processes will necessarily play a key role at elevations roughly above 1600 m asl., even in summer and especially in autumn; in summer, flash flood frequency will crucially depend on the saturation level of the catchment, which in turn is conditioned by the snowmelt season (duration, intensity); in autumn in exchange, potential early snowfall with or without subsequent rain-on-snow events can strongly modulate how intense precipitation events translate into flash floods or not.

  Regarding the modelling of antecedent moisture conditions please see the previous remarks. This is modelled and taken into account in the wflow_sbm model, Subsequently, this model is validated against historical flash floods and quite a few streamflow observations. Rain of snow events are not explicitly being modelled. We will note in the discussion that this a limitation of the current study.

- 2.6: daily maximal specific discharge for flash flood identification: why would the maximum discharge of a daily time step be relevant for small catchments?
  We take the maximal hourly discharge of each day for data processing considerations. This is not the same as the maximum discharge of a daily timestep. We will clarify the text.
- Flash flood definition: I see that only rainfall induced flash floods are considered; are rain-on-snow events never assimilated to flash floods? Given that the analysis expands on the alpine area, this should be made very clear; besides: are you sure that flash floods cannot occur early in spring (the analysis is on summer (JJA) and autumn (SON) period)? Especially in the future?
  Rain-on-snow events are not currently covered, this limitation of the study will be added to the discussion.
- Results: the fact that the he annual cycle of discharge with low flows in winter and snowmelt in May to June does not validate the hydrological model; any model will do so as long as there is any form of freezing and snow model in it; given the high annual cycle, we have no idea what the reported KGE values mean (any model that has some annual cycle will have KGE values beyond 0.6 I would guess); how could you validate the flood generation frequency of the modelling chain otherwise?
  We disagree to the statement that any model with some annual cycle will have KGE values beyond 0.6.
- Line 211: incomplete sentence (and update all figure references).
  Thank you for spotting this oversight.
- Discussion: I do not understand how you can judge if the frequency of events increases but at the same time admit that it is impossible to estimate return periods ("Just as in Rudd et al. (2020) we do not attempt to estimate return periods of extreme events as the simulation periods are too short to warrant such an approach."). I guess that answer is in Rudd et al. but it would be good to understand this here also.
  The procedure of estimating specific return periods for extreme events like floods, requires long timeseries (e.g. four times the time period of the desired return period, Tuyls et al., 2018). Although alternative statistical methods are being explored which aim to add validity to return periods estimated from short timeseries (e.g. Marra et al., 2018) like the 10 years in this study, the usefulness remains to be seen.

References:

Aerts, J. P. M., Hut, R. W., van de Giesen, N. C., Drost, N., van Verseveld, W. J., Weerts, A. H., and Hazenberg, P.: "Large-sample assessment of varying spatial resolution on the streamflow estimates of the wflow_sbm hydrological model", *Hydrology and Earth System Sciences,* 26, 4407–4430, https://doi.org/10.5194/hess-26-4407-2022 , 2022.

Amponsah, W., Ayral, P.-A., Boudevillain, B., Bouvier, C., Braud, I., Brunet, P., Delrieu, G., Didon-Lescot, J.-F., Gaume, E., Lebouc, L., et al.: Integrated high-resolution dataset of high-intensity European and Mediterranean flash floods, *Earth System Science Data*, 10, 1783–1794, 2018, https://doi.org/10.5194/essd-10-1783-2018

Bell, V. A., Kay, A. L., Jones, R. G., & Moore, R. J. (2007). "Development of a high resolution grid-based river flow model for use with regional climate model output". *Hydrology and Earth System Sciences*, 11(1), 532-549, https://doi.org/10.5194/hess-11-532-2007

N. Ban, C. Caillaud, E. Coppola, …, M.J. Zander, "The first multi-model ensemble of regional climate simulations at kilometer-scale resolution, part i: evaluation of precipitation." *Climate Dynamics*, pages 1–28, 2021. https://doi.org/10.1007/s00382-021-05708-w

Ségolène Berthou, Elizabeth J Kendon, Steven C Chan, Nikolina Ban, David Leutwyler, Christoph
Schär, and Giorgia Fosser. "Pan-European climate at convection-permitting scale: a model intercomparison study". *Climate Dynamics*, pages 1–25, 2018.
https://doi.org/10.1007/s00382-018-4114-6

Steven C Chan, Elizabeth J Kendon, Ségolène Berthou, Giorgia Fosser, Elizabeth Lewis, and Hayley J Fowler. "Europe-wide precipitation projections at convection permitting scale with the unified model". *Climate Dynamics*, 55(3):409–428, 2020.
https://doi.org/10.1007/s00382-020-05192-8

Froidevaux, P., Schwanbeck, J., Weingartner, R., Chevalier, C., and Martius, O.: Flood triggering in Switzerland: the role of daily to monthly preceding precipitation*, Hydrology and Earth System Sciences*, 19, 3903–3924, https://doi.org/10.5194/hess-19-3903-2015 , 2015.

R.O. Imhoff, W.J. van Verseveld, B. van Osnabrugge, and A.H. Weerts. "Scaling point-scale (pedo)transfer functions to seamless large-domain parameter estimates for high-resolution distributed hydrologic modeling: An example for the Rhine river". *Water Resources Research*, 56(4), April 2020. https://doi.org/10.1029/2019WR026807

Marra, F., Nikolopoulos, E. I., Anagnostou, E. N., and Morin, E.: Metastatistical Extreme Value analysis of hourly rainfall from short records: Estimation of high quantiles and impact of measurement errors, *Advances in Water Resources*, 117, 27–39, https://doi.org/https://doi.org/10.1016/j.advwatres.2018.05.001 , 2018.

Paprotny, D., Morales Napoles, O., and Jonkman, S. N.: HANZE database of historical damaging floods in Europe, 1870-2016., https://doi.org/10.4121/uuid:5b75be6a-4dd4-472e-9424-f7ac4f7367f6 , 2017.

Philippe Lucas-Picher, Daniel Argüeso, Erwan Brisson, Yves Tramblay, Peter Berg, Aude Lemonsu, Sven Kotlarski, and Cécile Caillaud. Convection-permitting modeling with regional climate models: Latest developments and next steps. *WIREs Climate Change*, 12 (6),e731 https://doi.org/10.1002/wcc.731

Emanuela Pichelli, Erika Coppola, Stefan Sobolowski, …, Jesus Vergara-Temprado. "The first multi-model ensemble of regional climate simulations at kilometer-scale resolution part 2: historical and future simulations of precipitation." *Climate Dynamics*, 56(11):3581–3602, 2021 https://doi.org/10.1007/s00382-021-05657-4

Alison C. Rudd, Alison L. Kay, Steven C. Wells, Timothy Aldridge, Steven J. Cole, Elizabeth J. Kendon, and Elizabeth J. Stewart. "Investigating potential future changes in surface water flooding hazard and impact". *Hydrological Processes*, 34(1):139–149, 2020. https://onlinelibrary.wiley.com/doi/abs/10.1002/hyp.13572

Schaller, N., Sillmann, J., Muller, M., Haarsma, R., Hazeleger, W., Hegdahl, T. J., Kelder, T., van den Oord, G., Weerts, A., and Whan, K. The role of spatial and temporal model resolution in a flood event storyline approach in western Norway, *Weather and Climate Extremes*, 29, 100 259, https://doi.org/https://doi.org/10.1016/j.wace.2020.100259 , 2020.

Frederiek C Sperna Weiland, Robrecht D Visser, Peter Greve, Berny Bisselink, Lukas Brunner, and Albrecht H Weerts. Estimating regionalized hydrological impacts of climate change over Europe by performance-based weighting of cordex projections. *Frontiers in Water*, 3:713537, 2021, https://doi.org/10.3389/frwa.2021.713537 .

Damian Murla Tuyls, Søren Thorndahl, Michael R. Rasmussen; Return period assessment of urban pluvial floods through modelling of rainfall–flood response. *Journal of Hydroinformatics* 1 July 2018; 20 (4): 829–845. doi: https://doi.org/10.2166/hydro.2018.133

W. J. van Verseveld, A. H. Weerts, M. Visser, J. Buitink, R. O. Imhoff, H. Boisgontier, L. Bouaziz, D. Eilander, M. Hegnauer, C. ten Velden, and B. Russell. "Wflow sbm v0.6.1, a spatially distributed hydrologic model: from global data to local applications". *Geoscientific Model Development Discussions*, 2022:1–52, 2022 (in review) https://doi.org/10.5194/gmd-2022-182

C. Wannasin, C.C. Brauer, R. Uijlenhoet, W.J. van Verseveld, A.H. Weerts, "Daily flow simulation in Thailand Part I: Testing a distributed hydrological model with seamless parameter maps based on global data", *Journal of Hydrology: Regional Studies*, 34, 2021, https://doi.org/10.1016/j.ejrh.2021.100794.